# ^1^H-NMR Based Serum Metabolomics Identifies Different Profile between Sarcopenia and Cancer Cachexia in Ageing Walker 256 Tumour-Bearing Rats

**DOI:** 10.3390/metabo10040161

**Published:** 2020-04-21

**Authors:** Laís Rosa Viana, Leisa Lopes-Aguiar, Rafaela Rossi Rosolen, Rogerio Willians dos Santos, Maria Cristina Cintra Gomes-Marcondes

**Affiliations:** Laboratory of Nutrition and Cancer, Department of Structural and Functional Biology, Biology Institute, University of Campinas (UNICAMP), Campinas 13083862, Brazil; lala.viana311088@gmail.com (L.R.V.); leisaaguiar@yahoo.com.br (L.L.-A.); rafaelarosolen@gmail.com (R.R.R.); rowisa@unicamp.br (R.W.d.S.)

**Keywords:** ageing, sarcopenia, cachexia, Walker 256 tumour

## Abstract

Sarcopenia among the older population has been growing over the last few years. In addition, the incidence of cancers increases with age and, consequently, the development of cachexia related cancer. Therefore, the elucidation of the metabolic derangements of sarcopenia and cachexia are important to improve the survival and life quality of cancer patients. We performed the ^1^H-NMR based serum metabolomics in adult (A) and ageing (S) Walker 256 tumour-bearing rats in different stages of tumour evolution, namely intermediated (Wi) and advanced (Wa). Among 52 serum metabolites that were identified, 21 were significantly increased in S and 14 and 19 decreased in the Wi and Wa groups, respectively. The most impacted pathways by this metabolic alteration were related by amino acid biosynthesis and metabolism, with an upregulation in S group and downregulation in Wi and Wa groups. Taken together, our results suggest that the increase in metabolic profile in ageing rats is associated with the higher muscle protein degradation that releases several metabolites, especially amino acids into the serum. On the other hand, we hypothesise that the majority of metabolites released by muscle catabolism are used by tumours to sustain rapid cell proliferation and tumorigenesis.

## 1. Introduction

Cancer-associated cachexia is a multifactorial syndrome characterised by the depletion of skeletal muscle and fat mass and cannot be fully reversed by conventional nutritional support [1]. Cachexia affects the majority of advanced cancer patients, especially those with lung, head and neck, gastro-oesophageal, and pancreatic cancers, and the main consequences are associated with progressive functional impairment, reduced physical performance, altered quality of life, poor prognosis and high mortality rates [1]. In addition, cachexia also occurs predominantly in elderly cancer patients and can worsen due to age-related physiologic loss of skeletal muscle mass and strength, a process known as sarcopenia [2,3].

The elucidation of the molecular mechanisms of sarcopenia and cachexia is important to prevent the process of skeletal muscle atrophy and, consequently, improve the survival of cancer patients [4,5]. In this context, many studies have been interested in identifying circulating markers of muscle wasting process during cancer cachexia in clinical [6,7,8,9] and preclinical [10,11,12] models.

Metabolomics studies have revealed a distinct plasma, serum, and urine metabolic profile of cancer cachexia patients [6,7]. Additionally, Boguszewicz and colleagues [8] described serum metabolic alterations as a characteristic for malnutrition or cachexia that can already be detected at the beginning of the treatment of head and neck squamous cell carcinoma patients. Miller and colleagues [9] identified candidate markers of weight loss in patients with upper gastrointestinal cancer and the assessment of the therapeutic intervention. However, it is important to mention that clinical data are limited because cachexia occurs at a stage in which patient vulnerability limits the use of invasive metabolic tests and disease progression limits the number of patients available for follow-up [1]. In this way, experimental cancer models can provide additional mechanistic insights.

The serum metabolic disturbances associated with enhanced citrate cycle and amino acid metabolism were the main characteristics of cachexia in tumour-bearing mice [10]. Furthermore, Lautaoja and collaborators [11] demonstrated that serum metabolomes are dysregulated in tumour-bearing mice, and suggested the free phenylalanine as a promising biomarker of muscle atrophy associated with cancer cachexia. Cancer-induced and chemotherapy-induced cachexia were characterised by metabolic derangements in mice, specifically in amino acid catabolism and influx through the tricarboxylic acid cycle and β-oxidation pathways [12].

Our previous studies have shown that the Walker 256 tumour growth—an experimental model of cachexia—led to metabolomic [13] and metabolic [14] alterations, and increased the muscle proteasomal activity [15,16] in adult rats. However, to the best of our knowledge, the biomarkers-related to sarcopenia and cachexia in senile cancer animals are not yet totally known. Therefore, we evaluated the tumour evolution effects on the alterations of serum metabolic profile and the main impacted metabolic pathways during the ageing process in Walker 256 tumour-bearing rats.

## 2. Results

### 2.1. Morphometric Parameters

Sarcopenia, imposed by the ageing process, led to a significant decrease in gastrocnemius muscle in the ageing control (S) group in comparison to the adult control (A) group. No reduction in gastrocnemius muscle tissue was found in intermediated (Wi) or advanced (Wa) tumour-bearing ageing groups when compared to S group. The liver tissue did not change in groups. Nevertheless, the body weight was significantly decreased in Wa in comparison to S group. As expected, the tumour weight was increased in Wa in comparison to Wi group showing an exponential evolution (Table 1).

### 2.2. Serum Metabolomic Profile

#### 2.2.1. Sarcopenia

Among 52 serum metabolites that were identified (Appendix A), 21 were significantly increased, and just one (dimethylamine) decreased in S in comparison to A group (Table 2). This alteration can be observed in the heatmap (Figure 1a) and the variable importance in projection (VIP) scores (Figure 1d). Both data clearly show the difference in the metabolomics concentration between S and A groups.

The most impacted pathways in S in comparison to A group by the metabolic alteration were the aminoacyl-tRNA biosynthesis (match status of 15 out of 48 metabolites); alanine, aspartate and glutamate metabolism (match status of 7 out of 48 metabolites); glyoxylate and dicarboxylate metabolism (match status of 6 out of 32 metabolites); glycine, serine and threonine metabolism (match status of 5 out of 34 metabolites); valine, leucine and isoleucine biosynthesis (match status of 3 out of 8 metabolites); arginine biosynthesis (match status of 3 out of 14 metabolites) and phenylalanine, tyrosine and tryptophan biosynthesis (match status of 2 out of 4 metabolites) (Table 3).

#### 2.2.2. Tumour Evolution

Among 52 serum metabolites that were identified (Appendix A), 14 and 19 metabolites were significantly decreased in the Wi and Wa groups, respectively, in comparison to S group (Table 2). This alteration can be observed in the heatmaps (Figure 1b,c) and VIP scores (Figure 1e,f).

The most impacted pathways by the metabolic alteration in Wi in comparison to S group were the aminoacyl-tRNA biosynthesis (match status of 9 out of 48 metabolites); alanine, aspartate and glutamate metabolism (match status of 6 out of 28 metabolites) and glyoxylate and dicarboxylate metabolism (match status of 5 out of 32 metabolites). And for Wa in comparison to S group the most impacted pathways were the aminoacyl-tRNA biosynthesis (match status of 13 out of 48 metabolites); alanine, aspartate and glutamate metabolism (match status of 8 out of 28 metabolites); glyoxylate and dicarboxylate metabolism (match status of 6 out of 32 metabolites); glycine, serine and threonine metabolism (match status of 4 out of 34 metabolites) and arginine biosynthesis (match status of 4 out of 14 metabolites) (Table 3).

No significant difference was observed in a comparison of serum metabolic profile between Wi and Wa groups (Appendix A).

## 3. Discussion

In this study, we aimed to study the muscle atrophy age-related process, termed as sarcopenia, and the changes in ageing tumour-bearing rats. The Walker 256 tumour experimental model was used to induce muscle atrophy related to tumour growth, termed as cancer cachexia.

As expected, sarcopenia decreased the gastrocnemius muscle mass according to other studies [17,18,19]. However, tumour evolution did not lead to a further reduction of gastrocnemius muscle mass but was associated with an accentuated decrease in body weight gain. Although cancer cachexia may also contribute to sarcopenia, the metabolic consequences and underlying pathophysiology differ in cachexia and in age-related muscle loss [20]. In cachexia, there is a decrease in the fat mass and an increase on muscle protein degradation and total energy expenditure, conditions that are not observed in sarcopenia [20,21,22].

In the context of better understanding the mechanisms that contribute to muscle wasting in sarcopenia and cancer cachexia, we performed a serum metabolic profile, using 1D ^1^H-NMR spectroscopy, comparing aged rats to the adult ones, and the negative effects of tumour evolution—intermediary and advanced growth, Wi and Wa groups—to the elderly group.

Among the total of 52 serum metabolites that were identified, 22 were changed between S and A groups, including an increase in asparagine, glutamate, leucine, lysine, methionine, proline, and valine that are involved in amino acid metabolism and biosynthesis pathways. Corroborating these results, Fazelzadeh and collaborators [23] observed that valine, leucine, proline, and methionine were reduced in the vastus lateralis muscle of healthy elderly patients in comparison to young subjects, suggesting that these metabolites could be increased in the serum. Moreover, a study performed by Calvani and colleagues [24] found that asparagine and glutamic acid increased in the serum of sarcopenic elderly than in non-sarcopenic elderly persons, these alterations suggested derangements in muscle energy metabolism associated with muscle wasting. This increase in the serum amino acid concentration could be related to the age-related muscle atrophy, but also might be related to the significant decline in the kidney function, usually associated with the ageing process. A study performed by Kouchiwa and colleagues [25] evaluated the age-related changes in serum amino acids concentrations in healthy individuals and found that serum amino acids concentrations are highly influenced by the ageing process. In according with this study, we also found an increase in amino acids serum concentrations such as asparagine, glutamate, alanine, tyrosine and phenylalanine in S group. In fact, the ageing process is well described that kidney system undergoes a variety of structural changes that leads to altered physiologic behaviour and, consequently impairs renal function [26,27]. This decline in kidney function could explain the general increase of most metabolites in serum due to a decreased filtration by the kidneys. This hypothesis of impaired renal function is further supported by a slight increase in serum creatinine and urea in S in comparison to A group. Serum creatinine and urea are standardly used for the evaluation of renal function in the clinic [28].

In order to evaluate if tumour growth would enhance this muscle metabolic alteration in sarcopenic rats, we analysed the serum metabolic profile of ageing tumour-bearing rats in different stages of tumour evolution, intermediated and advanced.

Indeed, the evolution of cancer led to important changes in the metabolomic profile. We observed here among the total of 52 serum metabolites identified a reduction in 14 metabolites in the intermediate growth of the tumour (Wi group). As a consequence of the severe tumour growth (Wa group), 19 metabolites decreased, being directly proportional to the serum metabolic alterations. According to our findings, the serum metabolomics clinical studies revealed decreased levels of carnitine, citrate, and valine in cancer cachexia [7], and also citrate and valine in prostate cancer [29] patients. Furthermore, low serum carnitine levels also contributed to the progression of cachexia in gastrointestinal cancer patients [30]. Also in line with our results, preclinical studies showed that the serum and plasma levels of citrate and valine were reduced in tumour-bearing mice [11,12] and serine decreased in Walker 256 tumour-bearing rats [13].

Additionally, it is important to emphasise that fumarate changed only in Wa group compared to non-tumour-bearing ageing rats. Since fumarate is a key molecule in the tricarboxylic cycle, especially in neoplastic cells, it is considered one of the oncometabolites, because is found in high concentrations in cancer and adjacent cells, suggesting that this metabolite could be decreased in the serum. This condition modifies the tumour microenvironment, sustaining the invasive neoplastic phenotype [31]. The role of fumarate as an oncometabolite could explain why this metabolite is the only one altered in Wa group, reinforcing the fact of higher metastatic process found in these animals (data not shown) and verified previously [13,14].

Moreover, an opposite metabolic profile was observed between sarcopenia and sarcopenia associated with cancer cachexia. The metabolic alterations in Wa group impacted mainly the aminoacyl-tRNA biosynthesis and alanine, aspartate and glutamate metabolism pathways, clearly related to the higher tumour cells activities. Similarly, Cala and colleagues [6] observed that these same pathways were impacted in cancer cachectic patients. In accordance with our results, Quan Jun and colleagues [32] also found in a cancer cachexia experimental model metabolite perturbations related to muscle atrophy. This metabolic derangement impacted some metabolic pathways such as serine and threonine metabolism; glyoxylate and dicarboxylate metabolism; alanine, aspartate and glutamate metabolism and arginine and proline metabolism.

Taken together, our results suggest that the increase in the metabolic profile in ageing rats is associated with the higher muscle protein degradation that releases several metabolites, especially amino acids into the serum, associated with a decrease kidney function, which likely explains the gastrocnemius muscle and body weight loss. On the other hand, we hypothesise as a severe effects that the majority of metabolites released by muscle catabolism are used by tumour to sustain the rapid cell proliferation and tumorigenesis, showing the inverse relation between tumour growth and body weight.

## 4. Materials and Methods

### 4.1. Animals

Adult Wistar rats (approximately 90 days old with 250 g of body weight) and ageing Wistar rats (approximately 400 days old with 650 g of body weight), obtained from the Animal Facilities at the State University of Campinas, UNICAMP, Brazil. All animals were housed in collective cages under controlled environmental conditions (light and dark 12/12 h; temperature 22  ±  2 °C, and humidity 50–60%). The general guidelines of the UKCCCR (United Kingdom Co-ordinating Committee on Cancer Research, 1998) regarding animal welfare were followed, and the experimental protocols were approved by the Institutional Committee for Ethics in Animal Research (CEEA/IB/UNICAMP, protocol # 3480-1; #5307-1).

### 4.2. Experimental Protocol

The animals were distributed into the following different experimental groups: adult control (A) (*n* = 4); ageing control (S) (*n* = 4) and ageing Walker 256 tumour-bearing rats (W), which were euthanised in different days of tumour evolution: approximately with 14 days, W intermediated (Wi) (*n* = 10), and approximately with 21 days, W advanced (Wa) (*n* = 10). The tumour implant was performed by a single subcutaneous inoculation of 2 × 106 viable neoplastic cells from Walker 256 tumour in the right flank of W groups. All animals were monitored daily, weighed three times/week, and received food and water ad libitum. The tumour dimensions (length, width and depth) were measured once a week using a slide calliper, to verify the properly tumour growth (data not shown). After each specific period of tumour evolution and after 21 days for both control groups (A and S), the rats were euthanised and the blood collected by cardiac puncture. Then, gastrocnemius muscle, liver and tumour tissue were weighed, and blood samples were centrifuged at 1000× *g* at 4 °C for 10 min, and serum was stored at −80 °C for metabolomic analyses.

### 4.3. Metabolomic Analyses

#### 4.3.1. Serum Preparation for ^1^H-NMR Acquisition

Serum samples were filtered through a Microcon YM-3 column (Amicon Ultra 0.5 mL, Sigma-Aldrich) with a 3-kDa membrane centrifuge filter for serum recovery (at 4 °C). Filtered serum (0.2 mL) was diluted in an aqueous solution (0.6 mL) containing 10% (*v*/*v*) deuterium oxide (D2O, 99.9%; Cambridge Isotope Laboratories Inc., Massachusetts, USA), phosphate buffer (0.1 M, pH 7.4) and 0.5 mM TMSP-d4 (3-(trimethylsilyl)-2,2’,3,3’-tetradeuteropropionic acid from Sigma-Aldrich) (internal reference), then transferred to a 5-mm nuclear magnetic resonance (NMR) tube (Norell Standard, Sigma-Aldrich, Missouri, United States of America) for immediate acquisition.

#### 4.3.2. H-NMR Spectra Acquisition and Metabolic Quantification

The 1D proton nuclear magnetic resonance (^1^H-NMR) spectra acquisition was performed using the Inova Agilent NMR spectrometer (Agilent Technologies Inc., Santa Clara, CA, USA) operating at a frequency of 600 MHz, equipped with a triple resonance cold probe at a constant temperature of 298 K (25 °C). A total of 256 scans were collected with 32-k data points over a spectral width of 8000 Hz. An acquisition time of 4 s and 1.5 s relaxation delay incorporated between scans [33], during which a continual water pre-saturation radiofrequency field was applied. After data acquisition, the spectroscopic data pre-processing was performed. Manual spectral processing, which included Fourier transform, phasing correction, baseline correction, water region deletion, shim correction, apodisation (line broadening with lb~0.3), and referencing control were applied before the profiling was performed. The identification and quantification of the metabolites were made by computer-assisted manual fitting using the Chenomx RMN Suite software (Chenomx Inc., Edmonton, Canada). To avoid bias, samples were randomly profiled blindly to the evaluator, and identified metabolites were fit to each spectrum by the same human operator, resulting in sample profiles consisting of each metabolite and its concentration is presented in millimoles per litre.

#### 4.3.3. Statistical Analysis

The morphometric parameters were presented as absolute and relative values. The delta body weight was calculated as (final body weight—initial body weight) for non-tumour-bearing groups and as ((final body weight-tumour weight)—initial body weight) for tumour-bearing groups. The relative values were calculated dividing gastrocnemius muscle, liver and tumour weights by the respective initial body weight of each animal. The data analyses were performed by one-way ANOVA, followed by post-hoc Tukey’s honestly significant difference (HSD) (comparison among A, S, Wi and Wa) and by *t*-test (comparison to Wa *versus* Wi). Data were expressed as mean ± standard deviation (SD) [34] and *P* < 0.05 was considered significant. The statistical analyses were performed using the software Graph Pad Prism 6.0 (Graph-Pad Software, Inc., San Diego, CA, USA).

The serum metabolic profile data analyses were performed by one-way ANOVA, followed by post-hoc Tukey’s HSD. Data were expressed as mean ± SD [33], and *P* < 0.05 was considered significant. The statistical analyses were performed using the software Graph Pad Prism 6.0 (Graph-Pad Software, Inc.).

The heatmaps and VIP scores data were expressed as abundance ratio (high and low) of the corresponding metabolite. The statistical analyses were performed using the online MetaboAnalyst 4.0 platform (a statistical, functional and integrative analysis of metabolomics data), more specifically in the “Statistical Analysis” tool [35].

The metabolic pathways were analysed with a list of metabolites—that were significantly different in the comparisons by one-way ANOVA, followed by post-hoc Tukey’s HSD—by over-representation analysis, using the hypergeometric tests (comparison to S *versus* A; Wi *versus* S; Wa *versus* S). Data were expressed as match status and regulation (upregulation or downregulation) and adjusted *P* < 0.01 by false discovery rate (FDR) was considered significant [36,37]. The statistical analyses were performed using the online MetaboAnalyst 4.0 platform (a statistical, functional and integrative analysis of metabolomics data), more specifically the “Pathway Analysis” tool [35,38].

## Figures and Tables

**Figure 1 metabolites-10-00161-f001:**
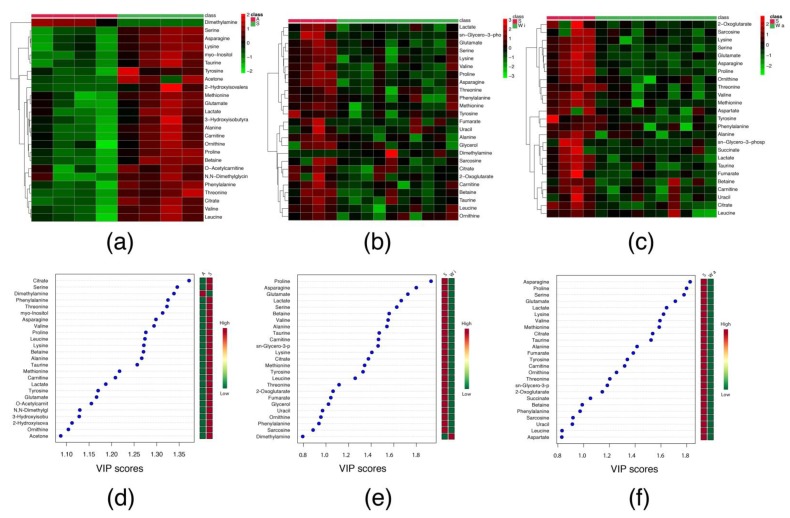
Heatmaps and Variable Importance in Projection scores of serum metabolic profile identified in adult, ageing and ageing Walker 256 tumour-bearing rats. (**a**–**c**) Heatmaps of comparison between S *versus* A; Wi *versus* S and Wa *versus* S, respectively; (**d**–**f**) Variable Importance in Projection (VIP) scores represents the comparison between S *versus* A; Wi *versus* S and Wa *versus* S, respectively. Legend: (A) control adult rat; (S) control ageing rat; (Wi) ageing tumour-bearing rat in intermediated stage; (Wa) ageing tumour-bearing rat in advanced stage. The red and green boxes indicate high and low abundance ratio, respectively, of the corresponding metabolite.

**Table 1 metabolites-10-00161-t001:** Morphometric parameters identified in adult, ageing and ageing Walker 256 tumour-bearing rats.

Morphometric Parameters	A		S		Wi		Wa	
	Mean ± SD (Absolute)	Mean ± SD (Relative)	Mean ± SD (Absolute)	Mean ± SD (Relative)	Mean ± SD (Absolute)	Mean ± SD (Relative)	Mean ± SD (Absolute)	Mean ± SD (Relative)
Initial body weight (g)	245.100 ± 33.310	-	673.100 ± 71.110	-	642.700 ± 81.380	-	598.900 ± 56.990	-
Final body weight (g)	314.100 ± 24.250	-	705.000 ± 71.110	-	624.200 ± 93.180	-	570.900 ± 100.400	-
Delta body weight (g)	75.000 ± 16.800	-	31.700 ± 12.100	-	−31.200 ± 31.020	-	−51.500 ± 74.550 ^b^	-
Gastrocnemius muscle weight (g)	2.194 ± 0.189	0.009 ± 0.001	2.780 ± 0.565	0.004 ± 0.001 ^a^	2.729 ± 0.900	0.004 ± 0.001	2.638 ± 0.884	0.004 ± 0.001
Liver weight (g)	9.442 ± 1.214	0.038 ± 0.003	18.770 ± 2.434	0.027 ± 0.004	15.710 ± 3.482	0.024 ± 0.005	15.080 ± 4.538	0.025 ± 0.007
Tumour weight (g)	-	-	-	-	12.600 ± 5.825	0.020 ± 0.010	23.560 ± 9.083	0.040 ± 0.017 ^c^

Legend: (A) control adult rat; (S) control ageing rat; (Wi) ageing tumour-bearing rat in intermediated stage; (Wa) ageing tumour-bearing rat in advanced stage. Delta body weight was calculated as [final body weight − initial body weight] for non-tumour-bearing groups and as [(final body weight − tumour weight) − initial body weight] for tumour-bearing groups. The relative values were calculated dividing gastrocnemius muscle, liver and tumour weights by the respective initial body weight of each animal. The results were presented as mean ± standard deviation (SD). Data were analysed by one-way ANOVA, followed by Tukey’s honestly significant difference (HSD) (comparison among A, S, Wi and Wa) and by t-test (comparison to Wa *versus* Wi). (a) *P*-value <0.05 in comparison to A group; (b) *P*-value <0.05 in comparison to S group and (c) *P*-value < 0.05 in comparison to Wi group.

**Table 2 metabolites-10-00161-t002:** Serum metabolic profile identified in adult, ageing and ageing Walker 256 tumour-bearing rats.

Metabolite	S *versus* A			Wi *versus* S			Wa *versus* S		
	Mean ± SD (mM)	Mean ± SD (mM)	*P*-Value	Mean ± SD (mM)	Mean ± SD (mM)	*P*-Value	Mean ± SD (mM)	Mean ± SD (mM)	*P*-Value
Alanine	0.245 ± 0.047	0.090 ± 0.049	**<0.001**	0.182 ± 0.037	0.245 ± 0.047	**0.041**	0.151 ± 0.029	0.245 ± 0.047	**0.001**
Asparagine	0.041 ± 0.007	0.016 ± 0.004	**<0.001**	0.020 ± 0.008	0.041 ± 0.007	**<0.001**	0.017 ± 0.004	0.041 ± 0.007	**<0.001**
Aspartate	0.019 ± 0.003	0.009 ± 0.005	**0.001**	0.013 ± 0.003	0.019 ± 0.003	**0.040**	0.010 ± 0.003	0.019 ± 0.003	**0.001**
Betaine	0.112 ± 0.027	0.035 ± 0.020	**0.005**	0.066 ± 0.024	0.112 ± 0.027	0.059	0.072 ± 0.036	0.112 ± 0.027	0.119
Carnitine	0.027 ± 0.006	0.010 ± 0.006	**0.001**	0.018 ± 0.004	0.027 ± 0.006	**0.019**	0.016 ± 0.005	0.027 ± 0.006	**0.006**
Citrate	0.147 ± 0.020	0.038 ± 0.020	**<0.001**	0.097 ± 0.036	0.147 ± 0.020	**0.044**	0.080 ± 0.029	0.147 ± 0.020	**0.005**
Dimethylamine	0.001 ± 0.0001	0.027 ± 0.008	**<0.001**	0.002 ± 0.002	0.001 ± 0.0001	0.946	0.002 ± 0.002	0.001 ± 0.0001	0.923
Fumarate	0.003 ± 0.002	0.002 ± 0.002	0.930	0.001 ± 0.001	0.003 ± 0.002	0.185	0.001 ± 0.0005	0.003 ± 0.002	**0.032**
Glutamate	0.087 ± 0.023	0.036 ± 0.022	**0.007**	0.056 ± 0.025	0.087 ± 0.023	0.063	0.037 ± 0.010	0.087 ± 0.023	**0.001**
Glutamine	0.374 ± 0.065	0.107 ± 0.061	**<0.001**	0.238 ± 0.066	0.374 ± 0.065	**0.014**	0.192 ± 0.076	0.374 ± 0.065	**0.001**
Glycine	0.141 ± 0.027	0.052 ± 0.027	**<0.001**	0.086 ± 0.017	0.141 ± 0.027	**0.008**	0.095 ± 0.032	0.141 ± 0.027	**0.029**
Lactate	5.588 ± 1.510	1.720 ± 1.541	**<0.001**	3.192 ± 0.856	5.588 ± 1.510	**0.002**	2.404 ± 0.499	5.588 ± 1.510	**<0.001**
Leucine	0.063 ± 0.007	0.027 ± 0.013	**0.005**	0.047 ± 0.011	0.063 ± 0.007	0.239	0.051 ± 0.017	0.063 ± 0.007	0.500
Lysine	0.163 ± 0.027	0.088 ± 0.056	**0.008**	0.117 ± 0.027	0.163 ± 0.027	0.069	0.093 ± 0.018	0.163 ± 0.027	**0.003**
Methionine	0.030 ± 0.004	0.014 ± 0.007	**<0.001**	0.021 ± 0.006	0.030 ± 0.004	**0.019**	0.018 ± 0.002	0.030 ± 0.004	**0.001**
Phenylalanine	0.034 ± 0.003	0.013 ± 0.006	**<0.001**	0.029 ± 0.007	0.034 ± 0.003	0.480	0.026 ± 0.004	0.034 ± 0.003	0.127
Proline	0.127 ± 0.025	0.046 ± 0.023	**<0.001**	0.062 ± 0.019	0.127 ± 0.025	**<0.001**	0.058 ± 0.011	0.127 ± 0.025	**<0.001**
Pyruvate	0.165 ± 0.022	0.022 ± 0.017	**<0.001**	0.102 ± 0.032	0.165 ± 0.022	**0.002**	0.100 ± 0.022	0.165 ± 0.022	**0.001**
Serine	0.124 ± 0.022	0.028 ± 0.019	**<0.001**	0.073 ± 0.020	0.124 ± 0.022	**<0.001**	0.060 ± 0.013	0.124 ± 0.022	**<0.001**
Taurine	0.304 ± 0.088	0.128 ± 0.082	**0.022**	0.170 ± 0.076	0.304 ± 0.088	**0.040**	0.148 ± 0.078	0.304 ± 0.088	**0.014**
Threonine	0.164 ± 0.034	0.052 ± 0.019	**0.002**	0.119 ± 0.037	0.164 ± 0.034	0.248	0.092 ± 0.047	0.164 ± 0.034	**0.024**
Tyrosine	0.049 ± 0.014	0.020 ± 0.009	**<0.001**	0.035 ± 0.007	0.049 ± 0.014	0.052	0.031 ± 0.007	0.049 ± 0.014	**0.011**
Valine	0.093 ± 0.014	0.039 ± 0.017	**<0.001**	0.062 ± 0.018	0.093 ± 0.014	**0.018**	0.052 ± 0.015	0.093 ± 0.014	**0.001**

Legend: (A) control adult rat; (S) control ageing rat; (Wi) ageing tumour-bearing rat in intermediated stage; (Wa) ageing tumour-bearing rat in advanced stage. Data were expressed as mean ± standard deviation (SD) and analysed by one-way ANOVA, followed by Tukey’s honestly significant difference (HSD) (comparison to S *versus* A; Wi *versus* S; Wa *versus* S). Bold *P* values represented a significant difference.

**Table 3 metabolites-10-00161-t003:** Metabolic pathways identified in adult, ageing and ageing Walker 256 tumour-bearing rats.

Pathway	Metabolite	S *versus* A			Wi *versus* S			Wa *versus* S		
		Match Status	Regulation	Adjusted *P*-Value *	Match Status	Regulation	Adjusted *P*-Value *	Match Status	Regulation	Adjusted *P*-Value *
Aminoacyl-tRNA biosynthesis	Alanine	15/48	↑	**<0.001**	9/48	↓	**<0.001**	13/48	↓	**<0.001**
	Asparagine		↑			↓			↓	
	Aspartate		↑			↓			↓	
	Glutamate		↑			ns			↓	
	Glutamine		↑			↓			↓	
	Glycine		↑			↓			↓	
	Leucine		↑			ns			ns	
	Lysine		↑			ns			↓	
	Methionine		↑			↓			↓	
	Phenylalanine		↑			ns			ns	
	Proline		↑			↓			↓	
	Serine		↑			↓			↓	
	Threonine		↑			ns			↓	
	Tyrosine		↑			ns			↓	
	Valine		↑			↓			↓	
Alanine, aspartate and glutamate metabolism	Alanine	7/48	↑	**<0.001**	6/28	↓	**<0.001**	8/28	↓	**<0.001**
	Asparagine		↑			↓			↓	
	Aspartate		↑			↓			↓	
	Citrate		↑			↓			↓	
	Fumarate		ns			ns			↓	
	Glutamate		↑			ns			↓	
	Glutamine		↑			↓			↓	
	Pyruvate		↑			↓			↓	
Glyoxylate and dicarboxylate metabolism	Citrate	6/32	↑	**<0.001**	5/32	↓	**<0.001**	6/32	↓	**<0.001**
	Glutamate		↑			ns			↓	
	Glutamine		↑			↓			↓	
	Glycine		↑			↓			↓	
	Pyruvate		↑			↓			↓	
	Serine		↑			↓			↓	
Glycine, serine and threonine metabolism	Betaine	5/34	↑	**0.001**	3/34	ns	0.054	4/34	ns	**0.011**
	Glycine		↑			↓			↓	
	Pyruvate		↑			↓			↓	
	Serine		↑			↓			↓	
	Threonine		↑			ns			↓	
Valine, leucine and isoleucine biosynthesis	Leucine	3/8	↑	**0.002**	1/8	ns	0.403	2/8	ns	0.033
	Threonine		↑			ns			↓	
	Valine		↑			↓			↓	
Arginine biosynthesis	Aspartate	3/14	↑	**0.012**	2/14	↓	0.095	4/14	↓	**<0.001**
	Fumarate		ns			ns			↓	
	Glutamate		↑			ns			↓	
	Glutamine		↑			↓			↓	
Phenylalanine, tyrosine and tryptophan biosynthesis	Phenylalanine	2/4	↑	**0.014**	0/4	ns	ns	1/4	ns	0.244
	Tyrosine		↑			ns			↓	

Legend: (A) control adult rat; (S) control ageing rat; (Wi) ageing tumour-bearing rat in intermediated stage; (Wa) ageing tumour-bearing rat in advanced stage. Data were represented as match status and regulation, which were analysed by over-representation analysis by hypergeometric tests (comparison to S *versus* A; Wi *versus* S; Wa *versus* S). More details see Material and Methods section. (↑) upregulation; (↓) downregulation; (ns) no significance. (*) adjusted *P*-value by false discovery rate (FDR) method. Bold *P* values represented a significant difference.

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
