# Peer review of "1H-NMR Based Serum Metabolomics Identifies Different Profile between Sarcopenia and Cancer Cachexia in Ageing Walker 256 Tumour-Bearing Rats"

_metabolites, 2020, doi:10.3390/metabo10040161_

Round 1

Reviewer 1 Report

In this manuscript, the authors conduct an experiment to compare several traits in adult and ageing rats in different stages of tumor evolution and studied the association with the metabolomics profiles. The goal of the study was to better understand the metabolic functions associated with sarcopenia and cachexia. Overall, the manuscript has its merits, but there are some concerns. The concerns and the suggestions are given below in order of their importance.

  1. There is lack of multiplicity adjustment in the results reported in Figure 1 and Table 1. For example, the authors seem to have corrected the p-values for the multiplicity across different metabolites, but they did not correct for multiplicity within a metabolite (Table 1) since they used Fisher post hoc test. This implies that they are doing 3 tests for each metabolite, but not correcting for that. This will lead to more false positives than what is acceptable. Although not clearly discussed, the reviewer believes that similar lack of adjustment also applies to the results in Table 2.

  1. The manuscript mentions “Mean pluminus SD” in several places. Do the authors mean “Mean plusminus SEM”? A simple cross check of the reported results makes the reviewer think that the what they report are indeed the standard error of mean. Please clarify. If they are indeed sample standard deviations, what is the purpose of reporting that? Also, it can be argued that, although it is common practice, reporting individual error bars (Mean plusminus SEM) in two-group comparisons is not very useful since what actually matters is the SE of the difference.

  1. Statistical analysis section should include more details of the pathway analysis and the PLS-DA. For example, what statistical method was used when the authors report p-values for the pathway analysis? The reviewer believes that simply referring to MetaboAnalyst is not enough.

  1. PLS-DA and VIP scores need reference when first mentioned (section 2.2.1).

  1. There are some typographical mistakes that should be carefully revised. Examples include:

     (i) Line 73: Incomplete sentence.

     (ii) Figure 2 caption: A, B, C, D, E, F instead of (a), (b), (c), (d), (e), (f).

  1. Are the commas in the data tables (e.g. Table S1) supposed to be decimal points? If not, why are the numbers formatted like that?

Author Response

Reviewer 1

Comments and Suggestions for Authors

In this manuscript, the authors conduct an experiment to compare several traits in adult and ageing rats in different stages of tumor evolution and studied the association with the metabolomics profiles. The goal of the study was to better understand the metabolic functions associated with sarcopenia and cachexia. Overall, the manuscript has its merits, but there are some concerns. The concerns and the suggestions are given below in order of their importance.

Answer: We would like to thank reviewer#1 for the time spent in the revision of our manuscript and for the valuable comments and suggestions. We are completely certain that the revised manuscript improved substantially after your suggestions.  The changes made in the revised paper are highlighted in yellow.

  1. There is lack of multiplicity adjustment in the results reported in Figure 1 and Table 1. For example, the authors seem to have corrected the p-values for the multiplicity across different metabolites, but they did not correct for multiplicity within a metabolite (Table 1) since they used Fisher post hoc test. This implies that they are doing 3 tests for each metabolite, but not correcting for that. This will lead to more false positives than what is acceptable. Although not clearly discussed, the reviewer believes that similar lack of adjustment also applies to the results in Table 2.

Answer: We agree that the statistical analyses in Figure 1 and Tables 1 and 2 were not well explained. We would like explaining in details that we have used one-way Anova and t-test (comparison between tumour-bearing groups) for morphometric analyses and t-test adjusted by False Discovery Rate (FDR) for all the metabolites analyses. In details, regarding to morphometric data (we are now showing all data of Figure 1 as a table format, now Table 1), we performed One-way ANOVA, followed by Fisher's least significant difference  (comparison among S, A, Wi and Wa) and t-test, when comparing both tumour-bearing groups (comparison to Wi versus Wa). For these parameters, P<0.05 was considered significant.

For analyses of serum metabolic profile, we performed t-test (comparison to S vs A; Wi vs S; Wa vs S) (now Table 2). For these parameters, the adjusted P<0.05 by False Discovery Rate (FDR) was considered significant. Furthermore, the metabolic pathways for these same comparations (old Table 2 and now table 3) were analyzed by over-representation analysis (hypergeometric tests) and considered adjusted P<0.05 by FDR. We considered the FDR for metabolomics data due to the multiple comparations performed (Benjamini and Hochberg 1995).

We also would like to inform that we have re-written the Statistical Analyses section with more details (page: 12; line: 242) and also better described the statistical analysis in the legends of Table1 (page 4; line: 102), Table 2 (page: 5; line: 108), Table 3 page: 9; line: 118), Supplementary Table1 (page: 16; line: 280) and Figure 1(page: 7; line: 111).

  1. The manuscript mentions “Mean pluminus SD” in several places. Do the authors mean “Mean plusminus SEM”? A simple cross check of the reported results makes the reviewer think that the what they report are indeed the standard error of mean. Please clarify. If they are indeed sample standard deviations, what is the purpose of reporting that? Also, it can be argued that, although it is common practice, reporting individual error bars (Mean plusminus SEM) in two-group comparisons is not very useful since what actually matters is the SE of the difference.

Answer: We have doubled-check our analyses, and as it is presented in the previous version we reported our results as mean±standard deviation (mean±SD) and not mean± standard error of mean (mean±SEM). The authors believe that the SD could be confused by SEM because of small SD in some cases, but this small value of SD is related to the lower metabolites concentration (e.g. for Dimethylamine between S vs A mean±SD= 0.001±0.0001 vs 0.027±0.008 and mean±SEM= 0.001±0.00004 vs 0.027±0.004). The purpose of reporting the SD is to show the variability (Altman and Bland 2005) among the different samples analysed, and we agree with the reviewer that reporting individual errors bars is not useful.

  1. Statistical analysis section should include more details of the pathway analysis and the PLS-DA. For example, what statistical method was used when the authors report p-values for the pathway analysis? The reviewer believes that simply referring to MetaboAnalyst is not enough.

Answer: We would like to thank the reviewer for raising the point that is lacking details of the methods on how we have made the pathway analysis. We excluded the PLS-DA analysis and now we are showing only the heatmaps as these analyses present enough informative differences among the groups.

With respect to the pathway analyses, we firstly made a list of metabolites that were significantly different (FDR <0.05) in the diverse comparations (S versus A: 22 metabolites; Wi versus S: 4 metabolites; Wa versus S: 14 metabolites. The list of specific metabolites can be found at Table 2). After that, the list of compounds was inserted in the MetaboAnalyst platform, more specifically in the Pathway Analysis tool (Xia and Wishart 2011). Data were analysed by over-representation analysis (hypergeometric tests) and considered as significant when adjusted P<0.05 by FDR. The hypergeometric test evaluates whether a particular metabolite set is represented more than expected by chance within the given compound list. The P-value indicates the probability of observing at least a particular number of metabolites from a certain metabolite set in a given compound list (Xia and Wishart 2011). We included in more details the description of metabolite and pathways analyses at Statistical Analysis section (page: 12; line: 242).

  1. PLS-DA and VIP scores need reference when first mentioned (section 2.2.1).

Answer: Thank you for raising this point. We included the VIP scores reference in the new manuscript version (page: 2; line: 80).

  1. There are some typographical mistakes that should be carefully revised. Examples include: (i) Line 73: Incomplete sentence; (ii) Figure 2 caption: A, B, C, D, E, F instead of (a), (b), (c), (d), (e), (f).

Answer: We revised the whole manuscript and specifically the typographical mistakes, and for this new manuscript version we have already checked and corrected this mistakes: (i) (page: 2; line: 71) and (ii) (page: 7; line: 111).

  1. Are the commas in the data tables (e.g. Table S1) supposed to be decimal points? If not, why are the numbers formatted like that?

Answer: Thank you for raising this point. We have now checked the whole manuscript and corrected and replaced commas by dots.

References:

  • Benjamini Y, Hochberg Y (1995) Controlling the false discovery rate: a practical and powerful approach to multiple testing. J Royal Stat Soc Ser B 57(1):449–518
  • Altman, D. G. and J. M. Bland (2005). "Standard deviations and standard errors." BMJ 331(7521): 903.
  • Xia, J. and D. S. Wishart (2011). "Web-based inference of biological patterns, functions and pathways from metabolomic data using MetaboAnalyst." Nat Protoc 6(6): 743-760.

Reviewer 2 Report

Review: 1H-NMR based serum metabolomics identifies different profile between sarcopenia and cancer cachexia in ageing Walker 256 tumour-bearing rats

by Laís Rosa Viana et al. 2020, submitted to Metabolites

General: In this study the authors compare the serum metabolic profiles of rats suffering either from sarcophenia or cancer cachexia. The paper is written in clear and concise English. However, as detailed below it has some serious issues with respect to both data acquisition and analysis that need to be addressed.

Specific:

p. 2 l. 68 The values of the morphometric parameters should be given in a table together with the information that normalized values are given. Furthermore, only relevant decimal places should be given.

p. 4 l. 101 table 1 Which statistical tests were used to identify significantly altered metabolites?

p. 6 l. 104 Figure 2. The heatmaps given are clearly informative. However, it is less clear why a PLS-DA was performed. As a PLS-DA includes as input information group-memberships it will in most cases give a good separation between groups, especially when the number of samples is relatively low compared to the number of features as it is here the case. Therefore, I would suggest to solely display the heatmaps.

p. 6 l. 113. I believe the authors talk here about the comparison between A and S group (please specify) and they discuss impacted pathways. However, the information provided how these pathways were identified is incomplete. Please specify which methods were exactly used. Furthermore, we look here at serum and the heatmap given in Figure 2A shows that most metabolites in the ageing group are increased compared to the adult group (see also table 2). Therefore, a likely explanation is that not specific metabolic pathways in the cells are altered but that what we see here is a decline in kidney function due to old age. This will lead to a general increase of most metabolites in blood due to a decreased filtration by the kidneys. This is further supported by the increase in serum creatinine in the ageing group (table S1). In my opinion the identified aminoacyl-tRNA biosynthesis pathway is an artifact due to the large number of regulated amino acids.

p 9 l. 176 The authors discuss the role of fumarate as an oncometabolite which is increased in cancer. However, the results show a decrease in fumarate for the rats with advanced tumor evolution, please comment. In addition, the authors state that fumarate is found in high concentrations in the tumor microenvironment. Do they really mean fumarate or is it confused with lactate? Furthermore, the comparison between Wa and S groups shows that metabolites are either downregulated or not regulated in the Wa group, indicating that metabolism in general is downregulated in the Wa group.

p. 10 l. 206 Please specify in the experimental protocol section the abbreviations used for the different groups of animals.

p. 11 l. 232 The authors state that 2k data points were collected over a spectral width of 8000 Hz. Please comment why such a small number of data points was collected as this will substantially decrease spectral resolution. Typically, 64 k or at least 32 k data points are collected. Please comment. Furthermore, the used relaxation delay was set to 1.5 seconds, which is clearly not sufficient to allow for complete relaxation of all compounds and thereby, will impact quantitative accuracy of obtained results.

p. 11 l. 252 The sentence “After that, different statistical analyses of the data were performed …” is rather unspecific. Please specify which tests were performed.

p. 12 l. 272 Table S1 please do an ANOVA for all metabolites given in the table and provide the corresponding p-values after correction for multiple testing independent whether they are significant or not.

Author Response

Reviewer 2

Comments and Suggestions for Authors

Review: 1H-NMR based serum metabolomics identifies different profile between sarcopenia and cancer cachexia in ageing Walker 256 tumour-bearing rats

General: In this study the authors compare the serum metabolic profiles of rats suffering either from sarcopenia or cancer cachexia. The paper is written in clear and concise English. However, as detailed below it has some serious issues with respect to both data acquisition and analysis that need to be addressed.

Answer: The authors would like to thank reviewer#2 for the time spent in the revision of our manuscript and for the valuable comments and suggestions. We are completely certain that the revised manuscript improved substantially after your suggestions.  The changes made in the revised paper are highlighted in yellow.

  1. 2 l. 68 The values of the morphometric parameters should be given in a table together with the information that normalized values are given. Furthermore, only relevant decimal places should be given.

Answer: We agree that the values of the morphometric parameters should be given in a table. The whole data of Figure 1 is now described in the new Table 1, which shows the absolute values of non-normalised data and the relative values which are normalised as the gastrocnemius muscle, liver and tumour weights were divided by the respective initial body weight of each animal. Delta body weight was calculated as: [final body weight - initial body weight] for non-tumour-bearing groups and as [(final body weight-tumour weight)-initial body weight] for tumour-bearing groups. We included this description in details in the main manuscript – section Material and Methods (page: 12; line: 242) and also in the legend description of Table 1 (page: 4; line: 102).

  1. 4 l. 101 table 1 Which statistical tests were used to identify significantly altered metabolites?

Answer: We have now included in the main text that we performed t-test (comparison to S vs A; Wi vs S; Wa vs S) (in previous version Table 1 and now Table 2). For these parameters adjusted P<0.05 by False Discovery Rate (FDR) was considered significant.

  1. 6 l. 104 Figure 2. The heatmaps given are clearly informative. However, it is less clear why a PLS-DA was performed. As a PLS-DA includes as input information group-memberships it will in most cases give a good separation between groups, especially when the number of samples is relatively low compared to the number of features as it is here the case. Therefore, I would suggest to solely display the heatmaps

Answer: We agree that the heatmaps are clearly informative and that PLS-DA was not providing any additional information and also not so clear as the heatmaps. As suggested, we solely presented the heatmaps. Please, see the new Figure 1 (page: 7, line: 111).

  1. 6 l. 113. I believe the authors talk here about the comparison between A and S group (please specify) and they discuss impacted pathways. However, the information provided how these pathways were identified is incomplete. Please specify which methods were exactly used.

Furthermore, we look here at serum and the heatmap given in Figure 2A shows that most metabolites in the ageing group are increased compared to the adult group (see also table 2). Therefore, a likely explanation is that not specific metabolic pathways in the cells are altered but that what we see here is a decline in kidney function due to old age. This will lead to a general increase of most metabolites in blood due to a decreased filtration by the kidneys. This is further supported by the increase in serum creatinine in the ageing group (table S1). In my opinion the identified aminoacyl-tRNA biosynthesis pathway is an artifact due to the large number of regulated amino acids.

Answer: We do thank reviewer for this important point that we were missing in our discussion. In the previous page 6 line 113 we were talking about the comparison between A and S group. We specified this comparison now in page 2, line 83. We agree that was lacking the information and description of the methods about we have made the identification of the impacted pathways in the comparison between S versus A group and also other comparisons such as [Wi vs S ] and [Wa vs S ]. So, we firstly made a list of metabolites that were significantly different (FDR <0.05) in the different comparations (S versus A: 22 metabolites; Wi versus S: 4 metabolites; Wa versus S: 14 metabolites. The list of specific metabolites can be found at Table 2). After that, this list of compounds was inserted in the MetaboAnalyst platform, more specifically in the pathway analysis toll (Xia and Wishart 2011). Data were analysed by over-representation analysis (hypergeometric tests). This hypergeometric test is used to evaluate whether a particular metabolite set is represented more than expected by chance within the given compound list. The P-value indicates the probability of observing at least a particular number of metabolites from a certain metabolite set in a given compound list (Xia and Wishart 2011). We included this description at Statistical Analysis section (page: 12; line: 242). Also, we would like to thank the reviewer for it valuable comment about the decline in kidney function associated with the ageing process. We agree that this fact should be considered in the interpretation of our results and we have now included a paragraph of discussion about kidney function. Please find the alteration in page: 10, line: 144. In fact, the serum creatinine tended to increase (S=0.016±0.002 vs A= 0.005±0.001; P-value=0.08; FDR=0.14) as well as the serum urea (S=0.639±0.076 vs A=0.198±0.056; P-value=0.07; FDR=0.13). We would like to emphasise that we have not included creatinine and urea in our Table 2 because of its FDR values, we have only considered as significant FDR < 0.05.

  1. p 9 l. 176 The authors discuss the role of fumarate as an oncometabolite which is increased in cancer. However, the results show a decrease in fumarate for the rats with advanced tumor evolution, please comment. In addition, the authors state that fumarate is found in high concentrations in the tumor microenvironment. Do they really mean fumarate or is it confused with lactate? Furthermore, the comparison between Wa and S groups shows that metabolites are either downregulated or not regulated in the Wa group, indicating that metabolism in general is downregulated in the Wa group.

Answer: We thank the reviewer for raising this point, but we really meant fumarate and its role as an oncometabolite. It is well established that fumarate is found in high concentrations in cancer and adjacent cells, modifying the tumour microenvironment and sustaining the invasive neoplastic phenotype (Dando, Pozza et al. 2019). Thus, we inferred that this could explain the decreased serum fumarate concentration since fumarate has a key role in invasive/metastatic phenotype and is probably mobilized by tumour at advanced stages. Furthermore, we believe that the general downregulated metabolism in Wa group, is the result of the nutrient mobilisation of tumour tissue growth. Different from sarcopenia (comparison between S and A groups), with an intense muscle degradation and also kidney dysfunction that lead to a general upregulation of metabolism, the tumour growth in older rats led to downregulated metabolism, favouring the tumour growth. Although we also included a discussion about the lactate and other metabolites that could infer an intense host catabolism associated to higher metabolic activity of tumour cells.

  1. 10 l. 206 Please specify in the experimental protocol section the abbreviations used for the different groups of animals.

Answer: We have now specified in the experimental protocol section the abbreviations used for the different groups of animals (page: 11, line: 203).

  1. 11 l. 232 The authors state that 2k data points were collected over a spectral width of 8000 Hz. Please comment why such a small number of data points was collected as this will substantially decrease spectral resolution. Typically, 64 k or at least 32 k data points are collected. Please comment. Furthermore, the used relaxation delay was set to 1.5 seconds, which is clearly not sufficient to allow for complete relaxation of all compounds and thereby, will impact quantitative accuracy of obtained results.

Answer: Thanks for raised this point. In fact, the number 3 is missing and actually 32 k data points were collected and not 2k. We have corrected this in the main text (page: 12; line: 230).

Regarding the relaxation/recycle delay, we applied similar NMR parameters that were used to build the metabolite library from Chenomx since we used this software to identify and quantify our NMR spectra. The Chenomx metabolite library was acquired with an acquisition time of 4 s and recycling delay of 1 s  (https://www.chenomx.com/support/). So, we used 1.5 s because it was even more that was suggested by Chenomx and also because some other recent studies performed in the same NMR applied these same parameters (Alborghetti, Correa et al. 2019, Castro, Duft et al. 2019). The information about the acquisition time was missing in the previous manuscript version and we have now included the acquisition time of 4 s in the main text (page: 12; line: 231).

  1. 11 l. 252 The sentence “After that, different statistical analyses of the data were performed …” is rather unspecific. Please specify which tests were performed.

Answer: We would like to inform that we have re-written the Statistical Analyses section with more details (page: 12; line: 242).

  1. 12 l. 272 Table S1 please do an ANOVA for all metabolites given in the table and provide the corresponding p-values after correction for multiple testing independent whether they are significant or not.

Answer: We have checked all data and reanalysed using one-way ANOVA for all metabolites given in the supplementary Table 1. Please find the new Table S1 at page: 14; line: 279. The statistical analysis is described at the Table legend.

References:

  • Alborghetti, M. R., M. E. P. Correa, J. Whangbo, X. Shi, J. A. Aricetti, A. A. da Silva, E. C. M. Miranda, M. L. Sforca, C. Caldana, R. E. Gerszten, J. Ritz and A. C. M. Zeri (2019). "Clinical Metabolomics Identifies Blood Serum Branched Chain Amino Acids as Potential Predictive Biomarkers for Chronic Graft vs. Host Disease." Front Oncol 9: 141.
  • Castro, A., R. G. Duft, M. L. V. Ferreira, A. L. L. Andrade, A. F. Gaspari, L. M. Silva, S. G. Oliveira-Nunes, C. R. Cavaglieri, S. Ghosh, C. Bouchard and M. P. T. Chacon-Mikahil (2019). "Association of skeletal muscle and serum metabolites with maximum power output gains in response to continuous endurance or high-intensity interval training programs: The TIMES study - A randomized controlled trial." PLoS One 14(2): e0212115.
  • Dando, I., E. D. Pozza, G. Ambrosini, M. Torrens-Mas, G. Butera, N. Mullappilly, R. Pacchiana, M. Palmieri and M. Donadelli (2019). "Oncometabolites in cancer aggressiveness and tumour repopulation." Biol Rev Camb Philos Soc 94(4): 1530-1546.
  • Xia, J. and D. S. Wishart (2011). "Web-based inference of biological patterns, functions and pathways from metabolomic data using MetaboAnalyst." Nat Protoc 6(6): 743-760.

Round 2

Reviewer 1 Report

The authors’ responses were mostly satisfactory, but there are still some concerns. The concerns and the suggestions are given below in order of their importance.

  1. I believe that the multiplicity issue is still not completely resolved. The authors have used Fisher’s LSD for post hoc comparisons after the ANOVA. Fisher’s LSD method does not adjust for multiple testing. I recommend using methods such as Tukey’s HSD.

    Also, the results from table 1 is unclear. The last sentence in the caption of table 1 seems incomplete: “(*) P-value <0.05 in comparison 105 to A group; (**) P-value <0.05 in comparison to S group; (***) P-value <0.05 in comparison to Wi group.” Are we comparing the Wa group here? Please clarify. Are the “*” used to imply strength of significance? If yes, for which variable? I suggest actually reporting the p-value like they were reported in the previous version. They can be reported either in the text or in table 1.  

  1. I still do not see any reference for the VIP scores (line 80).

  1. Line 71: “No reduction” instead of “Any reduction”.

  1. Duplication: Carefully revise lines 254-256, 261-263.

Author Response

Reviewer 1

Comments and Suggestions for Authors:

The authors’ responses were mostly satisfactory, but there are still some concerns. The concerns and the suggestions are given below in order of their importance.

            We would like to thank Reviewer 1 for the time spent in the revision of our manuscript. We are glad to know that the reviewer has considered most of our responses as satisfactory. Regarding the remaining concerns, we do believe that we could address them in this new version. The changes made in the revised paper are highlighted in yellow.

  1. I believe that the multiplicity issue is still not completely resolved. The authors have used Fisher’s LSD for post hoc comparisons after the ANOVA. Fisher’s LSD method does not adjust for multiple testing. I recommend using methods such as Tukey’s HSD.

            We agree with Reviewer 1 that the post-hoc Tukey HSD is a more adequate test to adjust for multiple testing. So, we re-analysed all data and we are now presenting data after being analysed by One-way ANOVA followed by post-hoc Tukey HSD. We have edit the data presented in Table 1 and Table 2. We also have changed the post-hoc test in the Statistical Analysis section (page:12; line: 291) and on Table 1 (page:4; line: 123), Table 2 (page:5; line: 129) and Table S1 legends (page:16; line: 338).

            For Table 1 data, now using the Tukey HSD for post hoc, the liver mass is not significantly different anymore between S vs A, as all as the delta body weight between S vs A; Wi vs S and Wa vs Wi. We are now reporting in Table 1 only the differences (P<0.05) found at Tukey HSD for post hoc, and these results did not change the main discussion and conclusion.

            After analysis of all metabolites using Tukey HSD post-hoc test (Table 2), we have found the same profile of the results presented in the previous version, although few metabolites that were significantly different in the previous version are not different anymore (S vs A: 3-Hydroxyisobutyrate; myo-Inositol; N,N-Dimethylglycine and O-Acetylcarnitine. Wi vs S: Glutamate) and some metabolites that were not different in the previous version are now significant (S vs A:Aspartate; Glutamine; Glycine and Pyruvate. Wi vs S: Alanine; Aspartate; Carnitine; Citrate; Glutamine; Glycine; Methionine; Pyruvate; Serine; Taurine and Valine. Wa vs S: Aspartate; Glutamine; Glycine; Pyruvate and Threonine).

            The list of the metabolites, with P<0.05 in the Tukey HSD post hoc, were used to re-analysed the impacted pathways. It is important to emphasise that all impacted pathway presented in the previous Table 3 were maintained as significant. Besides that, new pathways are now impacted by the new list of metabolites. In the previous version, the impacted pathways were:

  • Comparison between S and A groups: 1) Aminoacyl-tRNA biosynthesis; 2) Valine, leucine and isoleucine biosynthesis; 3) Alanine, aspartate and glutamate metabolism; 4) Glycine, serine and threonine metabolism and 5) Phenylalanine, tyrosine and tryptophan biosynthesis and now the impacted pathways are bolded as the additional via: 1) Aminoacyl-tRNA biosynthesis; 2) Alanine, aspartate and glutamate metabolism; 3) Glyoxylate and dicarboxylate metabolism; 4) Glycine, serine and threonine metabolism; 5) Valine, leucine and isoleucine biosynthesis; 6) Arginine biosynthesis and 7) Phenylalanine, tyrosine and tryptophan biosynthesis.
  • Comparison between Wi and S groups: 1) Aminoacyl-tRNA biosynthesis and now the impacted pathways are, in bold: 1) Aminoacyl-tRNA biosynthesis; 2) Alanine, aspartate and glutamate metabolism and 3) Glyoxylate and dicarboxylate metabolism.
  • Comparison between Wa and S groups: 1) Aminoacyl-tRNA biosynthesis and 2) Alanine, aspartate and glutamate metabolism and now the impacted pathways are, in bold: 1) Aminoacyl-tRNA biosynthesis; 2) Alanine, aspartate and glutamate metabolism; 3) Glyoxylate and dicarboxylate metabolism; 4) Glycine, serine and threonine metabolism and 5) Arginine biosynthesis.

            The pathways that are now presented as significant impacted is discussed in (page:10; line: 217). It is important to emphasise that these new pathways did not change the context of our discussion and conclusion. On the contrary, they reinforced and increased the impact of our results.

Also, the results from table 1 is unclear. The last sentence in the caption of table 1 seems incomplete: “(*) P-value <0.05 in comparison 105 to A group; (**) P-value <0.05 in comparison to S group; (***) P-value <0.05 in comparison to Wi group.” Are we comparing the Wa group here? Please clarify. Are the “*” used to imply strength of significance? If yes, for which variable? I suggest actually reporting the p-value like they were reported in the previous version. They can be reported either in the text or in table 1.

            We agree with the Reviewer 1 that the caption of Table 1 could be improved in order to clarify what we have made. The “*” previous used was not reporting the strength of significance but the comparison to A group. We have now replaced all * symbols to this description : (a) P-value <0.05 in comparison to A group; (b) P-value <0.05 in comparison to S group; (c) P-value <0.05 in comparison to Wi group (page:4; line: 126). We have compared the Wa group against the other groups, using the ANOVA followed by Tukey HSD post-hoc test, comparison among all groups.

  1. I still do not see any reference for the VIP scores (line 80).

            The reference for VIP scores is presented at Statistical Analyses section (page:13; line: 311). Reference number 35: Chong, J.; Wishart, D.S.; Xia, J. Using MetaboAnalyst 4.0 for comprehensive and integrative metabolomics data analysis. Curr Protoc Bioinformatics. 2019, 68, e86.

  1. Line 71: “No reduction” instead of “Any reduction”.

            We made this change in the main text. Please see (page:2; line: 80).

  1. Duplication: Carefully revise lines 254-256, 261-263.

            We would like to thank Reviewer 1 for raised this point. In fact, the 5-mm was duplicated in both sentences mentioned. We removed the “5-mm” from the second sentence. It was a typographical mistake.

Reviewer 2 Report

General: Most of my previous concerns were sufficiently addressed. There is only one small point remaining.

Specific:

p.12 l.259 As requested the authors state the test they were using for group comparisons. However, they used a standard t-test which is not adequate for multiple group comparisons, as it is here the case (table 2). For multiple group comparisons an ANOVA together with a post-hoc test as it was done for table S1 should be performed.

Author Response

Reviewer 2

Comments and Suggestions for Authors:

General: Most of my previous concerns were sufficiently addressed. There is only one small point remaining.

We would like to thank Reviewer 2 for the time spent in the revision of our manuscript. We are glad to know that the reviewer has considered that most of reviewer´s previous concerns were sufficiently addressed. Regarding the small point remaining, we do believe that we could address it in this new version. The changes made in the revised paper are highlighted in yellow.

Specific: p.12 l.259 As requested the authors state the test they were using for group comparisons. However, they used a standard t-test which is not adequate for multiple group comparisons, as it is here the case (table 2). For multiple group comparisons an ANOVA together with a post-hoc test as it was done for table S1 should be performed.

We agree with the Reviewer 2 that t-test is not adequate for multiple group comparisons. We are now presenting our data after being re-analysed by one-way ANOVA followed by the post-hoc Tukey HSD. We have edited the data presented in Table 1 and Table 2. We also have changed the post-hoc test in the Statistical Analysis section (page:12; line: 291) and on Table 1 (page:4; line: 123), Table 2 (page:5; line: 129) and Table S1 legends (page:16; line: 338).